# Signal Peptides and Their Fragments in Post-Translation: Novel Insights of Signal Peptides

**DOI:** 10.3390/ijms252413534

**Published:** 2024-12-18

**Authors:** Kenji Ono

**Affiliations:** 1Department of Neurotoxicology, Graduate School of Medical Sciences and Medical School, Nagoya City University, Nagoya 467-8601, Japan; k_ono@med.nagoya-cu.ac.jp; Tel.: +81-52-853-8992; Fax: +81-52-853-8996; 2Department of Brain Function, Division of Stress Adaptation and Protection, Research Institute of Environmental Medicine, Nagoya University, Nagoya 464-8601, Japan; 3Department of Molecular Pharmacokinetics, Nagoya University Graduate School of Medicine, Nagoya 464-8601, Japan

**Keywords:** signal peptides, extracellular vesicles, intercellular communication, post-translational function

## Abstract

Signal peptides (SPs), peptide sequences located at the N-terminus of newly synthesized proteins, are primarily known for their role in targeting proteins to the endoplasmic reticulum (ER). It has traditionally been assumed that cleaved SPs are rapidly degraded and digested near the ER. However, recent evidence has demonstrated that cleaved SP fragments can be detected in extracellular fluids such as blood flow, where they exhibit bioactivity. In addition, SP fragments are delivered to extracellular fluids via extracellular vesicles such as exosomes and microvesicles, which are important mediators of intercellular communication. These findings suggest that SPs and their fragments may have physiological roles beyond their classical function. This review aims to provide a comprehensive overview of these novel roles and offer new insights into the potential functions of SPs and their fragments in post-translational regulation and intercellular communication.

## 1. Introduction

SPs are peptide sequences located at the N-terminus of newly synthesized proteins, playing a crucial role as tags for targeting proteins to the ER during translation. SPs consist of three regions: a positively charged n-region, an h-region typically containing 8–20 hydrophobic residues, and a polar c-region, none of which show sequence conservation [1]. As proteins emerge from the ribosome, SPs direct them to the ER membrane through various targeting pathways and select different translocation systems for transport across the membrane [2]. Targeting and translocation can occur via signal recognition particle (SRP)-dependent or -independent pathways. In the SRP-dependent pathway, the SRP binds to the SP [3] and directs the ribosome to the ER translocon, arresting polypeptide elongation to ensure proper coordination between translation and protein location [4]. The SP is subsequently released from the SRP by its receptor on the ER, recognized by the translocon, and intercalated into the wall of the translocon [5]. The SRP-independent pathway also converges at the translocon [6]. The SP orientation is determined by the features of the SP sequence, with either the N-terminus or C-terminus remaining on the cytoplasmic side [7,8]. In type I orientation, the N-terminus of the SP faces the luminal side of the ER, whereas in type II orientation, it faces the cytoplasmic side. Protein synthesis resumes and SP-containing synthesizing proteins can pass into the ER membranes, where the SPs are cleaved from the synthesizing proteins by signal peptidases [9]. The membrane-bound SPs are further cleaved into two fragments by signal peptide peptidases [10]. It was previously thought that cleaved SPs are rapidly degraded and digested around the ER [11]. However, some SP fragments have been detected in extracellular fluids such as blood flow from both healthy individuals and patients, where they have demonstrated bioactivity [12]. This suggests that SP fragments are not always degraded intracellularly and may have additional physiological roles in post-translation. These discoveries have revealed new aspects of SPs and their fragments. SP fragments have shown potential as biomarkers for various diseases, and different SPs and their fragments from various cell types exhibit bioactivity. In certain cells, SP fragments are secreted via extracellular vesicles, which play important roles in intercellular communication. Furthermore, some diseases have been associated with SP abnormalities that are not related to ER targeting. These findings suggest that SPs may have unrecognized roles in physiological functions and disease mechanisms. Since the SP sequences differ between proteins and protein subclasses (Table 1), the functionality of each SP may also vary. Although some SPs and their fragments may have additional molecular and cellular functions beyond their classical role in ER targeting, the evidence is currently scattered and has not been systematically reviewed. Therefore, this review aims to provide new insights into SPs and their fragments and to explore their potential roles in post-translational processes.

## 2. Function of Signal Peptides

### 2.1. Classical Roles During Translation

SPs are located at the N-terminus of newly synthesized proteins and are essential for directing these proteins to the ER. Consequently, much research in molecular biology and biotechnology has focused on modifying SP sequences to alter protein expression levels [13]. By optimizing SP sequences, protein transport to the ER becomes more efficient, facilitating smooth protein folding and modification, which leads to increased production [14,15]. In addition, changes to SP sequences can affect ribosome binding efficiency and translation rates [16,17]. An optimal translation rate can enhance both the quality and quantity of proteins. Furthermore, optimized SPs improve secretion efficiency by promoting rapid cleavage and processing by signal peptidases [18,19]. The study of enhancing protein expression by modifying SP sequences is widely used as a key strategy to improve the efficiency and quality of protein production. This approach is expected to enable the mass production of biopharmaceuticals and industrial enzymes, thus improving cost efficiency and productivity.

### 2.2. Roles as Biomarkers

SPs involved in protein expression were once thought to be degraded and digested after cleavage by enzymes [11]. However, it is now known that at least some SP fragments are released into the extracellular fluid and possess biological activity. B-type natriuretic peptide (BNP) is a hormone that is synthesized and secreted by the heart (mainly in the ventricles), which acts to protect the myocardium through its diuretic, vasodilatory, renin-aldosterone secretory suppression, sympathetic suppression, and inhibition of hypertrophy [20,21,22]. Since BNP blood concentrations increase with higher cardiac workload and myocardial hypertrophy, it is widely used in clinical practice as a biomarker for heart failure, with BNP measurement useful for early detection of heart disease [23]. An SP fragment of BNP has been detected in the circulation of healthy individuals [12]. The full-length sequence of BNP SP consists of 26 amino acids, but the detected sequence was BNP_SP_ (17–26), a C-terminal fragment. Furthermore, BNP_SP_ (17–26) levels were elevated in the early stages of human myocardial infarction, compared to established markers such as NT-proBNP, myoglobin C, and troponin I [12], suggesting that BNP_SP_ (17–26) has potential as a novel biomarker of myocardial injury [24]. In addition, SP fragments of A-type natriuretic peptide (ANP) and C-type natriuretic peptide (CNP) have been found in human circulation [25,26] and are similarly expected to serve as biomarkers for cardiac diseases [25,27]. These findings suggest that SP fragments could be useful diagnostic markers. Furthermore, BNP_SP_ (17–26) has been shown to be protective in ischemia-reperfusion injury [28], suggesting that SP fragments may have potential not only as biomarkers but also for bioactive substances.

### 2.3. Post-Translational Roles

The first evidence that SPs have a post-translational role comes from antigen-presentation. Human Leukocyte Antigen-E (HLA-E) presents SP fragments of major histocompatibility complex (MHC) class I proteins on the cell surface [29,30]. MHC class I SP fragments increase HLA-E expression on the cell surface. Natural killer cells, which are capable of killing MHC class I-negative target cells, recognize HLA-E bound to SP fragments via CD94/NKG2 receptors, thereby inhibiting the lytic process [30,31,32]. In addition, HLA-E presents other SP fragments such as HSP60, leading to up-regulation HLA-E complexes on the cell surface of stressed cells [33]. However, this complex is not recognized by CD94/NGK2, and thus the lytic process is not inhibited. These findings indicate that SP fragment sequences possess bioactivity.

Recent studies have further clarified that several SPs function as bioactive peptides. Collagen, a major component of the extracellular matrix, acts as a cellular scaffold [34]. When SP fragments from ovalbumin, the most abundant protein in egg white, were added to mouse embryonic fibroblasts, changes in cell morphology and number were observed [35]. Further studies revealed that ovalbumin SP fragments suppressed integrin-mediated cell adhesion and focal adhesion formation, while modulating osteoblastic cell differentiation [36]. Eosinophil cationic protein (ECP), a toxin secreted by activated human eosinophils, is cytotoxic to lower organisms but not mammalian cells due to its cell membrane selectivity and ribonuclease activity. Interestingly, ECP’s N-terminal SP fragment also exhibits cytotoxicity and inhibits the growth of lower organisms [37]. This N-terminal SP fragment, cleaved by signal peptide peptidases, but not the C-terminal SP fragment, increased transforming growth factor-alpha and epidermal growth factor receptor expression at both mRNA and protein levels [38]. Moreover, the N-terminal SP fragment promotes the secretion of proinflammatory molecules, aiding macrophage migration to sites of inflammation [39]. G protein-coupled receptors (GPCRs) represent the largest protein family in vertebrates and play a crucial role in signal transduction [40]. GPCRs are membrane proteins with extracellular N-termini, and they utilize two distinct types of signal sequences. One group possesses SPs for ER targeting and insertion, which are cleaved by signal peptidases, while another group employs the first transmembrane domains as a non-cleavable signal anchor sequence in place of SPs. The former group constitutes a minority of GPCRs [41]. Protease-activated receptor 1 (PAR-1), a GPCR family member that mediates the thrombin signaling, has cleavable SPs [42]. PAR-1 SP has been shown to suppress ocular neovascularization and inflammation [43], as well as to contribute to cardiac and renal protection after ischemia and reperfusion injury [44,45,46]. AMPA-type glutamate receptors mediate fast excitatory neurotransmission and predominantly assemble as heterotetramers in the brain. The excisable SP of the AMPA receptor subunit GluA1 plays an unconventional role in regulating the spatial position of the subunit during heteromeric GluA1/A2 receptor assembly before being cleaved from the mature receptors. This suggests that glutamate receptor SPs may have additional cellular and molecular functions [47]. Kainate-type glutamate receptors, which are crucial for excitatory synaptic transmission and synaptic plasticity, are similarly regulated by SP fragments [48]. Surface trafficking and synaptic targeting of GluK1, a kainite-type glutamate receptor, are suppressed by cleaved GluK1 SPs, which directly interact with the N-terminal domain, acting as a ligand to suppress forward trafficking of the receptor [49]. These findings clearly demonstrate that SPs and their fragments perform diverse functions depending on their sequence.

## 3. Secretion of Signal Peptide Fragments via Extracellular Vesicles

### 3.1. Detection of Signal Peptide Fragments in Exosomes

One of the key questions has been why SPs, traditionally believed to undergo intracellular degradation, are detected in the extracellular fluid. Cells contain numerous enzymes, and the extracellular fluid also contains various enzymes, which are expected to be easily degraded and modified into peptides both intracellularly and extracellularly [50,51,52]. However, SPs and their fragments exhibit bioactivity, so it is necessary for them to function at the appropriate sites. Therefore, it has been postulated that mechanisms exist to prevent SP degradation and to transport them to their functional locations. Recently, extracellular vesicles have garnered attention for their important role in intercellular communication [53,54]. Extracellular vesicles, which carry bioactive substances such as miRNAs and proteins, can induce functional changes in recipient cells upon uptake. Two main types of extracellular vesicles are secreted by living cells: exosomes, which are formed and released via the endosomal pathway, and microvesicles, which are formed by shedding from the cell membrane [55]. Exosomes and microvesicles differ not only in their formation processes but also in their marker and functional molecules. For example, exosomes highly express specific markers such as tetraspanin families (CD63, CD9, and CD81) [56,57], heat shock proteins (HSP70 and HSP90) [58,59], and major histocompatibility class I molecules [60]. In contrast, microvesicles highly express cell surface antigens such as selectins [60,61] and integrins [62,63], as well as cytoskeletal proteins [64]. They also increase the membrane components of lipids, including cholesterol [65], phosphatidylserine [66], and diacylglycerol [67]. These compositional differences may lead to distinct roles for exosomes and microvesicles. The presence of SPs in extracellular vesicles like exosomes and microvesicles has led to the hypothesis that SP fragments might be secreted into the extracellular fluid. To investigate both the genetic activation and distribution of SPs, the SEAP (human secreted alkaline phosphatase) reporter system was used [68]. By replacing SEAP SP with specific SP sequences, gene expression was measured by SEAP activity in extracellular fluid, and the SP sequences were detected by mass spectrometry (Figure 1). A HEK293-based cell line (T-REx AspALP) was generated to produce APP (human amyloid precursor protein) SP instead of SEAP SP when expressing the SEAP reporter. The presence of APP SP fragments in extracellular vesicles was investigated [69]. APP SP fragments were detected in exosomes but not in microvesicles secreted by the T-REx AspALP cells. The detected fragment, APP_SP_ (10–17), spanned residues 10 to 17 at the C-terminal end, while residues 1 to 9 (APP_SP_ (1–9)) or the full-length sequence (APP_SP_ (1–17)) were not observed (Table 2). Since the APP SP is a type I SP, only the cytoplasmic side of the cleaved APP SP fragment was found in the exosomes. In contrast, the presence of SEAP SP, a type II SP, was investigated using HEK-Blue hTLR3 cells, which are HEK293 cells co-transfected with the *hTLR3* (human toll-like receptor 3) gene and the inducible *SEAP* reporter gene [70]. The SEAP SP fragment was detected in exosomes in the presence of poly(I:C), a TLR3 ligand, as the N-terminal fragment SEAP_SP_ (1–11), but not the C-terminal fragment and the full-length sequence (Table 2). These results indicate that exosomes contain the N- or C-terminal portion of SP fragments depending on the SP sequence, and these fragments represent the cytoplasmic side of cleaved SPs.

### 3.2. Detection of Signal Peptide Fragments in Microvesicles as Well as Exosomes

Like exosomes, microvesicles are also released from living cells [71]. While both microvesicles and exosomes encapsulate functional biomolecules such as proteins, miRNAs, and mRNAs, they differ in their formation processes and the abundance of their contents [72]. In addition, there is specificity in the cells that accepts these extracellular vesicles, which are thought to contribute to various forms of intercellular communication [73]. While it has been shown that SP fragments are released into the extracellular fluid via exosomes, it remains unclear whether SP fragments are released by microvesicles, another type of extracellular vesicle formed through a different process. Macrophages play an active role in intercellular communication mediated by extracellular vesicles. For example, exosomes released from the murine macrophage cell line RAW264.7 influence the proliferation and differentiation of mesenchymal stem cells [74], and macrophage-derived microvesicles have been reported to induce the differentiation of naive monocytes [75]. Since macrophages produce bioactive SPs, such as the CCL22 SP [76], it is likely that macrophage-derived SPs contribute to intercellular signaling mediated by extracellular vesicles. In response to stimuli such as lipopolysaccharide (LPS), a component of the outer membrane of Gram-negative bacteria, macrophages secrete numerous cytokines. Activated macrophages also respond to interferon-gamma (IFNγ), produced during both adaptive and innate immune responses, and tumor necrosis factor-alpha (TNFα) from antigen-presenting cells [77]. To investigate whether SP fragments are released into the extracellular fluid via macrophage-derived microvesicles, SEAP SP fragments were analyzed in RAW-Blue cells, which produce SEAP under the control of NF-κB in RAW264.7 cells in the presence or absence of activators such as LPS, TNFα, and IFNγ [78]. N-terminal fragments of SEAP SPs were detected in both microvesicles and exosomes, suggesting that SP fragment release into the extracellular fluid occurs via both types of extracellular vesicles, with distribution varying depending on the cell activation state.

### 3.3. Calmodulin’s Role in the Delivery of Signal Peptide Fragments to Extracellular Vesicles

The SP fragment of preprolactin has been reported to bind to calmodulin in a cell-free system [79]. Calmodulin, a Ca^2+^-binding protein, has a broad binding affinity for various peptide sequences [80]. It binds to elevated intracellular Ca^2+^ levels and acts as a Ca^2+^ buffer [81], interacting with various calmodulin-binding proteins to mediate physiological functions [82]. In the presence of Ca^2+^, the C-terminal fragment of APP SP was shown to bind calmodulin in a cell-free system, and this binding was enhanced with the addition of Ca^2+^ [70]. However, the binding was inhibited by the addition of a calmodulin inhibitory peptide [83]. These results suggest that SPs may bind to calmodulin intracellularly. Calmodulin has been reported to be detected in extracellular vesicles such as exosomes and microvesicles derived from various cell types, including mesenchymal stem cells, blood cells, and cancer cells [84,85,86]. Furthermore, when exosomes from T-REx AspALP cells were analyzed in the presence of W13, a calmodulin inhibitor [87], the levels of APP_SP_ (10–17) were detected in exosomes from W13-untreated cells but were reduced in exosomes from W13-treated cells [88]. Exosome-related proteins, such as Alix (ALG-2-interacting protein X) [89], CD63, and CD81, were also decreased in exosomes from W13-treated cells and increased in W13-treated cells. CD63 is localized in endosomes and exosomes [90], and the colocalization of CD63 and calmodulin was significantly observed in the W13-treated cells, indicating the accumulation of calmodulin in endosomes. In addition, the colocalization of CD63 and LAMP-2, a representative marker of lysosomes [91], was increased in the W13-treated cells, indicating increased degradation. When investigating what calmodulin was bound to inside the cells, it was found that in the absence of W13, calmodulin formed complexes with APP_SP_ (10–17), heat shock protein 70 (HSP70), and CD81. In contrast, in the presence of W13, no binding between calmodulin and APP_SP_ (10–17) was observed, while the binding levels of HSP70 and CD81 increased. Furthermore, the addition of EGTA, a calcium chelator, inhibited the increase. These results demonstrate that calmodulin plays an important role as a key regulator of the transport of SP fragments into exosomes. The detailed mechanism of how APP_SP_ (10–17), HSP70, and CD81 bind within the calmodulin complex remains unclear. HSP70 functions as a molecular chaperone and is involved in the prevention of protein aggregation and misfolding, trafficking, assembly, and degradation [92]. The binding of HSP70 and calmodulin is well known in the field of plant science [93,94,95] and their interaction is known to dissociate in the presence of EGTA [93]. HSP90, a member of the HSP family, binds to calmodulin and has multiple functions [96,97]. HSP70 interacts with other HSP families such as HSP90 [98], and HSP family members are also exosomal components, as is calmodulin [99,100]. On the other hand, CD81 is a marker for exosomes, as is CD63 and one of the tetraspanins, which are involved in numerous biological processes and modulate cell adhesion, motility, invasion, membrane fusion, signal transduction, and protein trafficking [101,102,103]. It remains unclear whether calmodulin directly binds to CD81, but calmodulin has been reported to bind to other tetraspanins like peripherin/rds, found in photoreceptor cells [104]. Thus, it is possible that calmodulin interacts with tetraspanins including CD81. The interaction between calmodulin, heat shock proteins, and tetraspanins may play a crucial role in the transport of SP fragments to exosomes.

In addition, SEAP SP content in both microvesicles and exosomes decreases in the presence of a calmodulin inhibitor [78], further supporting calmodulin’s role in SP transport to extracellular vesicles. These findings suggest that the cleaved SP fragments on the ER may bind to calmodulin and be transported to exosomes or microvesicles depending on the cell’s activation state (Figure 2). Calmodulin is involved in multiple intracellular signaling pathways, and its role in SP fragment translocation likely varies based on the cell’s signaling state and activation conditions [105,106]. Given the bioactivity of SP fragments, the functional differences between exosomes and microvesicles could be partly explained by differences in their SP fragment content.

## 4. Diseases Associated with Signal Peptide Abnormalities

Abnormalities in the functions and properties of SPs are associated with various diseases. Mutations in SPs affect protein targeting, translocation, processing, and stability [107]. For example, SP mutations in genes such as *CRB1*, *NDP*, *FZD4*, *EYS*, and *RS1* are rare causes of inherited retinal diseases [108]. A mutation of the preproparathyroid hormone gene has been proposed as a cause of familial isolated hypoparathyroidism, with a point mutation in the SP coding region identified in one family [109]. This mutation disrupted the hydrophobic core of the SP, which is necessary for efficient translocation of secreted proteins across the ER, leading to improper processing of the mutant protein. During skin desquamation, kallikrein-related peptidases (KLKs) played a key role by degrading corneodesmosomes. The KLK11 Gly50Glu variant caused an autosomal dominant cornification disorder by subcellular mislocalization and secretion deficiency, resulting from impaired SP cleavage [110]. Most of these diseases appear to stem from abnormalities in the classical functions of SPs during translation, but some are due to issues in post-translational SP processing that lead to protein accumulation. Recent studies suggest that SPs may also be involved in the accumulation of beta-amyloid (Ab), a hallmark of Alzheimer’s disease [111,112]. APP_SP_ (1–17), an SP fragment of the APP, forms amyloid-like aggregates and enhances Ab aggregation due to the hydrophobic sequence, which induces cytotoxicity. In addition, the SP of ADAM17 SP has been shown to form amyloid-like aggregates under buffered conditions, and these aggregates interact with Ab peptides. Typically, SP sequences consist of three regions, with the central region being hydrophobic. In these studies, the hydrophobic region was found to contribute to the aggregation of both SPs and Ab peptides. In an analysis of extracellular vesicles from T-REx AspALP cells, where APP_SP_ (1–17) was expressed in place of the original SP, only APP_SP_ (10–17) was detected in the extracellular vesicles, while APP_SP_ (1–17) was not [34]. Furthermore, APP_SP_ (1–17) was not found in the ER fraction of these cells. These findings suggest that APP_SP_ (1–17) is generally cleaved into two fragments by signal peptide peptidases. Therefore, dysfunction of signal peptide peptidases could potentially lead to the aggregation of APP SP and Ab. Interestingly, engineered cell-penetrating SPs have been shown to inhibit Ab aggregation and reduce its cytotoxicity [113,114]. However, when the sequences of these engineered SPs were altered, they sometimes promoted Ab aggregation and cytotoxicity. These observations indicate that intracellular SPs may interfere with Ab aggregation, possibly through extracellular vesicles acting as regulatory factors. In addition, exosomes have been shown to cross the BBB in healthy and diseased conditions, although the mechanism is not fully understood [115]. Analysis of SPs in extracellular vesicles could be used as biomarkers for disease diagnosis. Therapeutic applications based on the bioactivity of SPs in extracellular vesicles may also become possible. Understanding the role of SPs in extracellular vesicles could provide new insights into the treatment of neurodegenerative diseases.

## 5. Future Perspectives and Conclusions

Extracellular vesicles contain functional molecules such as miRNAs and proteins that selectively alter the functions of recipient cells. Recent studies have shown that SPs also act as functional molecules (Table 3). Therefore, SP fragments within extracellular vesicles are believed to function as biologically active substances in recipient cells. Future studies should focus on elucidating the functions and biological significance of SPs and their fragments in extracellular vesicles. Since SP sequences vary depending on the protein and its subclass (Table 1), if SPs served solely as targeting sequences to the ER, it would be reasonable to expect them to share common sequences. However, the variability suggests that different SPs may play distinct roles. To fully understand the function of SP fragments, it is essential to investigate the biological activities of individual SPs and accumulate the results. Just as proteome analysis using mass spectrometry has contributed significantly to the analysis of proteins contained in extracellular vesicles, signal peptidome analysis of peptides extracted from extracellular vesicles may be essential for research development. In addition, it is necessary to create extracellular vesicles that contain specific SP fragments and extracellular vesicles that do not, and to study the changes in the properties of the cells that are individually taken up.

Some SPs detected in extracellular fluids such as blood are considered potential biomarkers [12,25,26]. SP fragments have been shown to be released into the extracellular fluids via extracellular vesicles such as exosomes and microvesicles [69,78]. Since calmodulin has been found to bind to SPs [70,88], it may be possible to enhance biomarker detection by enriching and analyzing extracellular vesicles and calmodulin complexes from blood for peptides.

There is also the possibility that SP fragments found in extracellular vesicles and extracellular fluid are derived from dead cells. However, in experiments using the SEAP reporter system, no changes in cell viability or increases in dead cells were observed under various stimulations, suggesting that SP fragments do not originate from dead cells [69,70]. Moreover, when cells were stimulated, the abundance and orientation of SP fragments in extracellular vesicles changed [78], suggesting that SP fragments in extracellular fluids are not merely accidental products of digested cellular contents. One possibility is that SP fragments contribute to intercellular communication, as they exhibit bioactivity similar to that of miRNAs and proteins in extracellular vesicles. Another possibility is that SP fragments produced by excessive protein synthesis are exported from the intracellular environment to extracellular fluid. Most unnecessary proteins are degraded intracellularly via the ubiquitin-proteasome system [116], and proteasomal degradation typically results in small peptides that are further hydrolyzed into amino acids. However, some peptides generated by proteasomes are presented on MHC class I molecules. These findings suggest that even if excess SP fragments are released via extracellular vesicles, they may still contribute to intercellular communication, signaling cellular conditions to neighboring cells.

Calmodulin plays a crucial role in transporting SP fragments to extracellular vesicles [88]. Calmodulin is a calcium-binding protein with four EF-hand motifs, and it changes conformation depending on calcium concentrations [117], increasing its binding affinity for SPs in the presence of calcium ions [70]. Calmodulin is currently the only known protein that binds to the SPs, but other molecules may also be involved. Calcium-binding proteins like calpain [118,119] and ALG-2 (apoptosis linked gene-2) [120] possess five EF-hand motifs. ALG-2 is a component of the ESCRT (Endosomal Sorting Complex Required for Transport) system [121], which is involved in exosome formation. ALG-2 plays a role in exosome biosynthesis by interacting with exosome-associated proteins such as Alix and Tsg101 (tumor susceptibility gene 101) [122,123,124]. In addition, ALG-2 has Ca^2+^-dependent membrane-binding properties [120] and is involved in plasma membrane repair via interactions with ESCRT-related proteins [125]. Like calmodulin, ALG-2 binds to a variety of proteins and peptides and may act as an adaptor for SP transport to extracellular vesicles. Further studies should explore other potential adaptor molecules involved in SP transport.

In conclusion, SPs are not merely ER-targeting sequences during translation. In some cases, SPs may act before cleavage, but in many cases, cleaved SP fragments have been shown to be biologically active and perform various functions. SP fragments have been found within extracellular vesicles, which are crucial for signal transduction between specific cells and are released into the extracellular fluids, where they may contribute to previously unrecognized intercellular interactions in recipient cells. Since SP sequences are not conserved even among homologous species, it is necessary to analyze the functions of each peptide individually. In addition, some diseases are associated with SP abnormalities, which may result from a lack of SP cleavage rather than from protein synthesis defects during translation. Understanding the role of SPs and their fragments could provide important insights into their physiological significance.

## Figures and Tables

**Figure 1 ijms-25-13534-f001:**
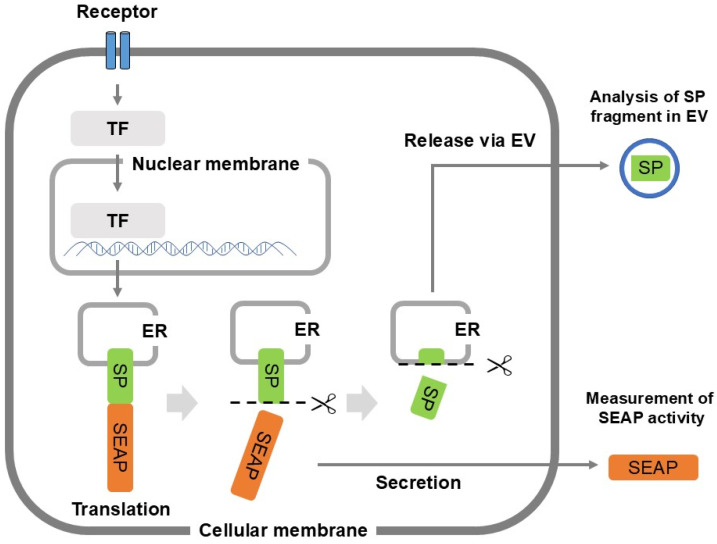
Detection of SP fragment using SEAP reporter system. SEAP SP was exchanged for specified SP, and specified SP and SEAP reporter sequences were stably expressed in clonal cells. Based on gene expression, SEAP activity in extracellular fluid was measured for translational activity. In addition, distribution of SP and their fragments in the ER and the EV were examined with a mass spectrometry TF; transcription factor, EV; extracellular vesicle, enzymes such as signal peptidase and signal peptide peptidases indicated as an icon of scissors.

**Figure 2 ijms-25-13534-f002:**
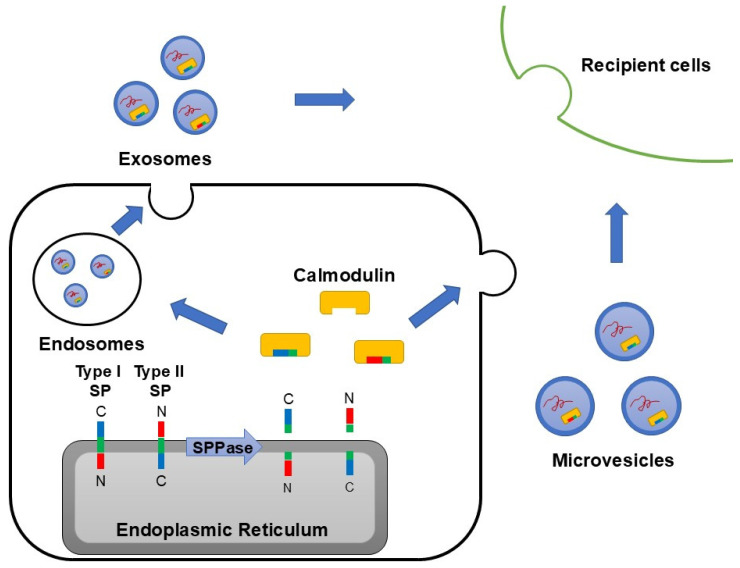
Secretion of SP fragments via extracellular vesicles such as exosomes and microvesicles. The cytoplasmic side of SP fragments cleaved by signal peptide peptidase (SPPase) was bound to calmodulin and distributed in exosomes and microvesicles. Since SP fragments had physiological functions and extracellular vesicles played important roles in selective intercellular communication, it would be very important to clarify the role of SP fragments in recipient cells in future studies.

**Table 1 ijms-25-13534-t001:** SP sequences of human cathepsin subclasses. Confirmed SP sequences of cathepsins were retrieved from the Signal Peptide Database (http://www.signalpeptide.de/, accessed on 27 June 2024). Each cathepsin subclass has different SP sequences.

Accession Number	Protein Name	Signal Sequence
P07339	Cathepsin D	MQPSSLLPLALCLLAAPA
P14091	Cathepsin E	MKTLLLLLLVLLELGEAQG
P08311	Cathepsin G	MQPLLLLLAFLLPTGAEA
P09668	Cathepsin H	MWATLPLLCAGAWLLGVPVCGA
P56202	Cathepsin W	MALTAHPSCLLALLVAGLAQG

**Table 2 ijms-25-13534-t002:** Sequences of APP and SEAP SP and their fragments.

Name	Signal Sequence
APP_SP_ (1–17)	_1_MLPGLALLLLAAWTARA_17_
APP_SP_ (1–9)	_1_MLPGLALLL_9_
APP_SP_ (10–17)	_10_LAAWTARA_17_
SEAP_SP_ (1–17)	_1_MLLLLLLLGLRLQLSLG_17_
SEAP_SP_ (1–11)	_1_MLLLLLLLGLR_11_
SEAP_SP_ (12–17)	_12_LQLSLG_17_

**Table 3 ijms-25-13534-t003:** Roles of SPs and their fragments in post-translation.

Category	SP Name	Roles	References
Biomarker	A-type natriuretic peptide	Biomarker of cardiac diseases	[25]
	B-type natriuretic peptide	Biomarker of myocardial injury	[12,24]
	C-type natriuretic peptide	Biomarker of cardiac diseases	[27]
Bioactivity	B-type natriuretic peptide	Protection from ischemia and reperfusion injury	[28]
	Ovalbumin	Cell adhesion and differentiation	[36]
	Eosinophil cationic protein	Cytotoxicity of lower organisms	[37]
		Expression of TGFβ and EGFR	[38]
		Secretion of proinflammatory molecules	[39]
	Protease-activated receptor 1	Suppression of ocular neovascularization and inflammation	[43]
		Protection from ischemia and reperfusion injury	[44,45,46]
	GluK1 kainate receptor	Inhibition of the synaptic and surface expression of GluK1	[49]
Transport via extracellular vesicles	Amyloid precursor protein	Release into the extracellular fluid via exosomes	[69,88]
	Secreted alkaline phosphatase	Release into the extracellular fluid via exosomes and microvesicles	[70,78]

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
