# Peer review of "Signal Peptides and Their Fragments in Post-Translation: Novel Insights of Signal Peptides"

_ijms, 2024, doi:10.3390/ijms252413534_

Round 1

Reviewer 1 Report

Comments and Suggestions for Authors

L 35 - L 36

Please briefly describe the function or reason for SRP arrests or delays elongation of the nascent chain during targeting.

L 63 

If possible, add a column that briefly summarizes the functionality of the different SPs.

L 71

What strategies are generally used to optimize SP sequences?

L 102 - L 103

Is there a level/concentration/dose difference between BNP as a biomarker or bioactive substance?

L 120 - L 123

How do mammalian cells survive or grow unaffected when treated with ECP?

L 164

What factors determine the different modes of secretion of different SPs (Exosomes or microvesicles)?

L 194

Is it possible to add a column that reviews how they are secreted? Or where they are detected?

L 228

Any studies on the potential mechanisms of this dynamic balance of activation states and secretion patterns?

L 276 - L 277

Calmodulin can also have many targets on individual vesicles, so it remains to be determined exactly how calmodulin regulates each of the membrane traffic steps in which it is involved.

L 319 - L 321

The effect of peptide crossing the blood-brain barrier should be further reviewed.

Author Response

I wish to express my appreciation to you for your insightful comments on my review paper. The comments have helped me significantly improve the paper. I have highlighted the revision parts (yellow line).

L 35 - L 36

Please briefly describe the function or reason for SRP arrests or delays elongation of the nascent chain during targeting.

Thank you for your precise suggestion. As you know, arresting elongation serves several critical functions such as prevention of misfolding and aggregation, efficient targeting to the ER membrane, and synchronization with translocation machinery. Therefore, I briefly added the sentence “ to ensure proper coordination between translation and protein location.” 

L 63 

If possible, add a column that briefly summarizes the functionality of the different SPs.

Unfortunately, it is difficult to add a column for the signal peptide of the cathepsin because its function is not known except that it is targeted to the ER.

L 71

What strategies are generally used to optimize SP sequences?

The optimization of SP sequences is crucial for improving the efficiency of protein secretion, especially in biotechnological and industrial applications. therefore, a variety of strategies are used. The strategies include codon optimization, signal peptide selection, hydrophobic core modification,  length optimization, cleavage efficiency improvement, contextual optimization, synthetic and computational design, and toxicity avoidance. This paper focused on the novel roles of SP, so the details were not written.

L 102 - L 103

Is there a level/concentration/dose difference between BNP as a biomarker or bioactive substance?

BNP is a clinically established biomarker widely used in the diagnosis and monitoring of heart failure and cardiovascular disease. On the other hand, BNP SP is expected to complement BNP as a potential new biomarker for early myocardial stress and specific pathological mechanisms. This point was briefly explained in Lines 89-97.

L 120 - L 123

How do mammalian cells survive or grow unaffected when treated with ECP?

Thank you for your suggestion. For the brief explanation, I added the sentence “due to its cell membrane selectivity and ribonuclease activity” in Line 123-124.

L 164

What factors determine the different modes of secretion of different SPs (Exosomes or microvesicles)?

Thank you for your important comments. As indicated in this review, it is clear that SPs are transported in both exosomes and microvesicles, that their orientation changes depending on the activation state of the cell, and that the binding of calmodulin and SP fragments is important for both, but the detailed molecular mechanisms remain unclear. I believe that this needs to be clarified in future studies.

L 194

Is it possible to add a column that reviews how they are secreted? Or where they are detected?

Thank you for your suggestion. I made Table 3.

L 228

Any studies on the potential mechanisms of this dynamic balance of activation states and secretion patterns?

Reference 78 shows changes in the orientation of signal peptides to exosomes and microvesicles when states of NF-kB activation via different signaling pathways are established. It was also found that in both systems the signal peptide is bound to and transported by calmodulin. Future studies will further elucidate the mechanism.

L 276 - L 277

Calmodulin can also have many targets on individual vesicles, so it remains to be determined exactly how calmodulin regulates each of the membrane traffic steps in which it is involved.

Thank you for your important comments. Future studies will further elucidate the detailed mechanism.

L 319 - L 321

The effect of peptide crossing the blood-brain barrier should be further reviewed.

Thank you for your suggestion. I added the sentence “In addition, exosomes have been shown to cross the BBB in healthy and diseased conditions, although the mechanism is not fully understood [115].” in Line 320-Line 322.

List of modifications

Line 2

Modifying “Signal peptide” to “Signal peptides”

Line 36-37

Adding the sentence "to ensure proper coordination between translation and protein location"

Line 70-71

Modification of the sentence to “SPs are located at the N-terminus of newly synthesized proteins and are essential for directing these proteins to the ER.”

Line 123-124

Adding the sentence “due to its cell membrane selectivity and ribonuclease activity”

Line 176

Modifying “human secreted alkaline phosphate” to “human secreted alkaline phosphatase”

Line 320-322

Adding the sentence “In addition, exosomes have been shown to cross the BBB in healthy and diseased conditions, although the mechanism is not fully understood [115].”

Line 322-324

Adding the sentences “Analysis of SPs in extracellular vesicles could be used as biomarkers for disease diagnosis. Therapeutic applications based on the bioactivity of SPs in extracellular vesicles may also become possible.”.

Line 330

Adding the sentence “(Table 3)”.

Line 338-343

Adding the sentences “Just as proteome analysis using mass spectrometry has contributed significantly to the analysis of proteins contained in extracellular vesicles, signal peptidome analysis of peptides extracted from extracellular vesicles may be essential for research development. In addition, it is necessary to create extracellular vesicles that contain specific SP fragments and extracellular vesicles that do not, and to study the changes in the properties of the cells that are individually taken up.”

Line 345-347

Adding Table3 and the legend.

Line 365

Modifying “[115]” to ”[116]”.

Line 373

Modifying “[116]” to ”[117]”

Line 376

Modifying “[117,118]” to ”[118,119]” and “[119]” to “[120]”.

Line 378

Modifying “[120]” to ”[121]”

Line 380

Modifying “[121-123]” to ”[122-124]”

Line 381

Modifying “[119]” to ”[120]”

Line382

Modifying “[124]” to ”[125]”

Line 400

Modifying “Ab” to “Ab”

Line 413

Modifying “IFNg” to “IFNg”

Line 416

Modifying “NF-kB” to “NF-kB”

Line 418

Modifying “human secreted alkaline phosphate” to “human secreted alkaline phosphatase”

Line 421

Modifying “TNFa” to “TNFa”

Line 505

Modifying “Wu, C.-M.; Chang, M.D.-T.” to “Wu, C.M.; Chang, M.D.T.”

Line 510

Modifying “Liu, Y.-S.; Tsai, P.-W.; Wang, Y.; Fan, T.; Hsieh, C.-H.; Chang, M.D.-T.; Pai, T.-W.; Huang, C.-F.; Lan, C.-Y.; Chang, H.-T.” to “Liu, Y.S.; Tsai, P.W.; Wang, Y.; Fan, T.; Hsieh, C.H.; Chang, M.D.T.; Pai, T.W.; Huang, C.F.; Lan, C.Y.; Chang, H.T.”

Line 512

Modifying “BMC Systems Biology 2012 6:1” to “BMC Systems Biology

Line 513

Modifying “EMBO J 1999,” to “EMBO J 1999,”

Line 536

Modifying “Duan, G.-F” to “Duan, G.F.”

Line 539

Modifying “J Biol Chem 2008” to J Biol Chem 2008

Line 544

Modifying “Cells 2022, 11,” to “Cells 2022, 11, 1375,”

Line 595

Modifying “PeerJ 2020, 2020, e8970” to “PeerJ 2020, 8, e8970”

Line 688-689

Adding the reference “Abdelsalam, M.; Ahmed, M.; Osaid, Z.; Hamoudi, R.; Harati, R. Insights into Exosome Transport through the Blood–Brain Barrier and the Potential Therapeutical Applications in Brain Diseases. Pharmaceuticals 2023, 16, 571, doi:10.3390/PH16040571.”

Line 690-713

Changing the reference number.

Reviewer 2 Report

Comments and Suggestions for Authors

The author has prepared a review of the literature regarding noncanonical functions of signal peptides. This is an interesting topic, and a somewhat understudied aspect of molecular and cellular biology. The author is comprehensive in their review of the alternative roles that SPs have been known to fill, and these are each addressed in detail. Overall, I find this review to the thoughtfully and carefully crafted.  In addition, the author provides excellent interpretation and analysis of prior work, with perspectives for future avenues of research included as well.

Comments on the Quality of English Language

The quality of the English is mostly very good, but there are several places where a few minor improvements would increase the clarity of the work. There are a few redundancies (for instance, the first sentence of sections 1 and 2 are nearly identical).  This could easily be remedied with minor revisions.

Author Response

I wish to express my appreciation to you for your insightful comments on my review paper. The comments have helped me significantly improve the paper. I have highlighted the revision parts (yellow line).

There are a few redundancies (for instance, the first sentence of sections 1 and 2 are nearly identical).  This could easily be remedied with minor revisions.

Thank you for your suggestion. I modified the sentence in Line 69-70 to “SPs are located at the N-terminus of newly synthesized proteins and are essential for directing these proteins to the ER.”

List of modifications

Line 2

Modifying “Signal peptide” to “Signal peptides”

Line 36-37

Adding the sentence "to ensure proper coordination between translation and protein location"

Line 70-71

Modification of the sentence to “SPs are located at the N-terminus of newly synthesized proteins and are essential for directing these proteins to the ER.”

Line 123-124

Adding the sentence “due to its cell membrane selectivity and ribonuclease activity”

Line 176

Modifying “human secreted alkaline phosphate” to “human secreted alkaline phosphatase”

Line 320-322

Adding the sentence “In addition, exosomes have been shown to cross the BBB in healthy and diseased conditions, although the mechanism is not fully understood [115].”

Line 322-324

Adding the sentences “Analysis of SPs in extracellular vesicles could be used as biomarkers for disease diagnosis. Therapeutic applications based on the bioactivity of SPs in extracellular vesicles may also become possible.”.

Line 330

Adding the sentence “(Table 3)”.

Line 338-343

Adding the sentences “Just as proteome analysis using mass spectrometry has contributed significantly to the analysis of proteins contained in extracellular vesicles, signal peptidome analysis of peptides extracted from extracellular vesicles may be essential for research development. In addition, it is necessary to create extracellular vesicles that contain specific SP fragments and extracellular vesicles that do not, and to study the changes in the properties of the cells that are individually taken up.”

Line 345-347

Adding Table3 and the legend.

Line 365

Modifying “[115]” to ”[116]”.

Line 373

Modifying “[116]” to ”[117]”

Line 376

Modifying “[117,118]” to ”[118,119]” and “[119]” to “[120]”.

Line 378

Modifying “[120]” to ”[121]”

Line 380

Modifying “[121-123]” to ”[122-124]”

Line 381

Modifying “[119]” to ”[120]”

Line382

Modifying “[124]” to ”[125]”

Line 400

Modifying “Ab” to “Aβ”

Line 413

Modifying “IFNg” to “IFNγ”

Line 416

Modifying “NF-kB” to “NF-κB”

Line 418

Modifying “human secreted alkaline phosphate” to “human secreted alkaline phosphatase”

Line 421

Modifying “TNFa” to “TNFα”

Line 505

Modifying “Wu, C.-M.; Chang, M.D.-T.” to “Wu, C.M.; Chang, M.D.T.”

Line 510

Modifying “Liu, Y.-S.; Tsai, P.-W.; Wang, Y.; Fan, T.; Hsieh, C.-H.; Chang, M.D.-T.; Pai, T.-W.; Huang, C.-F.; Lan, C.-Y.; Chang, H.-T.” to “Liu, Y.S.; Tsai, P.W.; Wang, Y.; Fan, T.; Hsieh, C.H.; Chang, M.D.T.; Pai, T.W.; Huang, C.F.; Lan, C.Y.; Chang, H.T.”

Line 512

Modifying “BMC Systems Biology 2012 6:1” to “BMC Systems Biology

Line 513

Modifying “EMBO J 1999,” to “EMBO J 1999,”

Line 536

Modifying “Duan, G.-F” to “Duan, G.F.”

Line 539

Modifying “J Biol Chem 2008” to J Biol Chem 2008

Line 544

Modifying “Cells 2022, 11,” to “Cells 2022, 11, 1375,”

Line 595

Modifying “PeerJ 2020, 2020, e8970” to “PeerJ 2020, 8, e8970”

Line 688-689

Adding the reference “Abdelsalam, M.; Ahmed, M.; Osaid, Z.; Hamoudi, R.; Harati, R. Insights into Exosome Transport through the Blood–Brain Barrier and the Potential Therapeutical Applications in Brain Diseases. Pharmaceuticals 2023, 16, 571, doi:10.3390/PH16040571.”

Line 690-713

Changing the reference number.

Reviewer 3 Report

Comments and Suggestions for Authors

Summary: 

The manuscript reviews the novel roles of signal peptides (SPs) and their fragments beyond their classical function in targeting proteins to the endoplasmic reticulum (ER). It explores their bioactivity in extracellular fluids, potential as biomarkers, and role in intercellular communication via extracellular vesicles such as exosomes and microvesicles. The review highlights the biological significance of SPs, their release mechanisms, and their association with diseases. Key insights include SPs' role in post-translational regulation, their bioactivity in the extracellular environment, and the role of calmodulin in transporting SP fragments to extracellular vesicle. 

Suggestions:

1. The author introduced various functions of SPs, however, it would be beneficial to expand on the molecular mechanisms underlying these processes, especially regarding calmodulin-mediated transport.

2. The author discussed diseases related to SP abnormalities but should elaborate on therapeutic implications, particularly how engineered SPs could be used in treatment or diagnosis.

3. Expand the “Future Perspectives” section by detailing specific experimental approaches to validate the proposed roles of SP fragments, such as proteomics for vesicle content analysis or functional assays in recipient cells.

4. Include more recent references (2020-2024) to provide updated perspectives on SP research.

5. It would be helpful to add a table summarizing SP functions across various contexts (e.g., biomarker roles, extracellular vesicle transport).

Author Response

I wish to express my appreciation to you for your insightful comments on my review paper. The comments have helped me significantly improve the paper. I have highlighted the revision parts (yellow line).

The author introduced various functions of SPs, however, it would be beneficial to expand on the molecular mechanisms underlying these processes, especially regarding calmodulin-mediated transport.

Thank you for your suggestion. It is clear that SPs are transported in both exosomes and microvesicles, that their orientation changes depending on the activation state of the cell, and that the binding of calmodulin and SP fragments is important for both, but the detailed molecular mechanisms remain unclear. Future studies will further elucidate the detailed mechanism.

The author discussed diseases related to SP abnormalities but should elaborate on therapeutic implications, particularly how engineered SPs could be used in treatment or diagnosis.

Thank you for your suggestion. I added the sentences “Analysis of SPs in extracellular vesicles could be used as biomarkers for disease diagnosis. Therapeutic applications based on the bioactivity of SPs in extracellular vesicles may also become possible.” in Line 322-324.

Expand the “Future Perspectives” section by detailing specific experimental approaches to validate the proposed roles of SP fragments, such as proteomics for vesicle content analysis or functional assays in recipient cells.

Thank you for your suggestion. I added the sentences “Just as proteome analysis using mass spectrometry has contributed significantly to the analysis of proteins contained in extracellular vesicles, signal peptidome analysis of peptides extracted from extracellular vesicles may be essential for research development. In addition, it is necessary to create extracellular vesicles that contain specific SP fragments and extracellular vesicles that do not, and to study the changes in the properties of the cells that are individually taken up.” in Line 338-343.

Include more recent references (2020-2024) to provide updated perspectives on SP research.

I added a recent reference as Reference 115.

It would be helpful to add a table summarizing SP functions across various contexts (e.g., biomarker roles, extracellular vesicle transport).

I added a table summarizing SP roles as Table 3.

List of modifications

Line 2

Modifying “Signal peptide” to “Signal peptides”

Line 36-37

Adding the sentence "to ensure proper coordination between translation and protein location"

Line 70-71

Modification of the sentence to “SPs are located at the N-terminus of newly synthesized proteins and are essential for directing these proteins to the ER.”

Line 123-124

Adding the sentence “due to its cell membrane selectivity and ribonuclease activity”

Line 176

Modifying “human secreted alkaline phosphate” to “human secreted alkaline phosphatase”

Line 320-322

Adding the sentence “In addition, exosomes have been shown to cross the BBB in healthy and diseased conditions, although the mechanism is not fully understood [115].”

Line 322-324

Adding the sentences “Analysis of SPs in extracellular vesicles could be used as biomarkers for disease diagnosis. Therapeutic applications based on the bioactivity of SPs in extracellular vesicles may also become possible.”.

Line 330

Adding the sentence “(Table 3)”.

Line 338-343

Adding the sentences “Just as proteome analysis using mass spectrometry has contributed significantly to the analysis of proteins contained in extracellular vesicles, signal peptidome analysis of peptides extracted from extracellular vesicles may be essential for research development. In addition, it is necessary to create extracellular vesicles that contain specific SP fragments and extracellular vesicles that do not, and to study the changes in the properties of the cells that are individually taken up.”

Line 345-347

Adding Table3 and the legend.

Line 365

Modifying “[115]” to ”[116]”.

Line 373

Modifying “[116]” to ”[117]”

Line 376

Modifying “[117,118]” to ”[118,119]” and “[119]” to “[120]”.

Line 378

Modifying “[120]” to ”[121]”

Line 380

Modifying “[121-123]” to ”[122-124]”

Line 381

Modifying “[119]” to ”[120]”

Line382

Modifying “[124]” to ”[125]”

Line 400

Modifying “Ab” to “Aβ”

Line 413

Modifying “IFNg” to “IFNγ”

Line 416

Modifying “NF-kB” to “NF-κB”

Line 418

Modifying “human secreted alkaline phosphate” to “human secreted alkaline phosphatase”

Line 421

Modifying “TNFa” to “TNFα”

Line 505

Modifying “Wu, C.-M.; Chang, M.D.-T.” to “Wu, C.M.; Chang, M.D.T.”

Line 510

Modifying “Liu, Y.-S.; Tsai, P.-W.; Wang, Y.; Fan, T.; Hsieh, C.-H.; Chang, M.D.-T.; Pai, T.-W.; Huang, C.-F.; Lan, C.-Y.; Chang, H.-T.” to “Liu, Y.S.; Tsai, P.W.; Wang, Y.; Fan, T.; Hsieh, C.H.; Chang, M.D.T.; Pai, T.W.; Huang, C.F.; Lan, C.Y.; Chang, H.T.”

Line 512

Modifying “BMC Systems Biology 2012 6:1” to “BMC Systems Biology

Line 513

Modifying “EMBO J 1999,” to “EMBO J 1999,”

Line 536

Modifying “Duan, G.-F” to “Duan, G.F.”

Line 539

Modifying “J Biol Chem 2008” to J Biol Chem 2008

Line 544

Modifying “Cells 2022, 11,” to “Cells 2022, 11, 1375,”

Line 595

Modifying “PeerJ 2020, 2020, e8970” to “PeerJ 2020, 8, e8970”

Line 688-689

Adding the reference “Abdelsalam, M.; Ahmed, M.; Osaid, Z.; Hamoudi, R.; Harati, R. Insights into Exosome Transport through the Blood–Brain Barrier and the Potential Therapeutical Applications in Brain Diseases. Pharmaceuticals 2023, 16, 571, doi:10.3390/PH16040571.”

Line 690-713

Changing the reference number.